# Genome-Wide Identification and Expression Profiling of Dehydration-Responsive Element-Binding Family Genes in Flax (*Linum usitatissimum* L.)

**DOI:** 10.3390/ijms26073074

**Published:** 2025-03-27

**Authors:** Yan Wang, Yanni Qi, Limin Wang, Chenmeng Xu, Wenjuan Li, Zhao Dang, Wei Zhao, Ping Wang, Yaping Xie, Yamin Niu, Nan Lu, Zuyu Hu, Zigang Liu, Jianping Zhang

**Affiliations:** 1College of Agriculture, Gansu Agricultural University, Lanzhou 730070, China; 1073323120598@st.gsau.edu.cn (Y.W.); 1073324120574@st.gsau.edu.cn (Y.N.); 1073324120393@st.gsau.edu.cn (N.L.); 1073324120594@st.gsau.edu.cn (Z.H.); lzgworking@163.com (Z.L.); 2Institute of Crop, Gansu Academy of Agricultural Sciences, Lanzhou 730070, China; xbsdqyn@126.com (Y.Q.); liminwang@aliyun.com (L.W.); 18735906104@163.com (C.X.); liwenjuan@gsagr.ac.cn (W.L.); dangzhao@gsagr.cn (Z.D.); zhaowei@gsagr.ac.cn (W.Z.); wangping1976@gsagr.cn (P.W.); xieyp2012@126.com (Y.X.); 3State Key Laboratory of Aridland Crop Science, Lanzhou 730070, China

**Keywords:** flax, *DREB*, transcription factors, abiotic stress, expression analysis

## Abstract

Dehydration-responsive element-binding (*DREB*) transcription factors are ubiquitous in plants and regulate plant growth, development, signal transduction, and responses to stress, particularly drought stress. However, *DREB* genes in flax have not previously been studied. This study conducted a comprehensive and systematic analysis of the *DREB* gene family in flax (*Linum usitatissimum* L.). A total of 59 *LuDREB* genes were identified in Longya-10 (a breeding variety), with an uneven distribution across all 15 chromosomes. Further analysis revealed significant variations among LuDREB members, with predictions indicating that these proteins are hydrophilic and localized in the nucleus and cytoplasm. A phylogenetic analysis classified the *LuDREB* genes into six subgroups, a classification further supported by gene structure and motif composition. Members within the same subgroup exhibited structural conservation, suggesting functional redundancy. The duplication analysis identified 30 pairs of segmentally duplicated *LuDREB* genes and one pair of tandemly duplicated genes, indicating that segmental duplication was the primary driver of *LuDREB* gene expansion. A comparative collinearity analysis revealed that most *LuDREB* genes had orthologs in other plant species, suggesting that this gene family has remained relatively conserved throughout evolution. *Cis*-acting element analysis identified numerous hormone- and stress-responsive elements in *LuDREB* promoters, and the quantitative real-time reverse transcription polymerase chain reaction (qRT-PCR) results confirmed the role of all *LuDREB* genes in drought stress response. In addition, transcriptome analysis revealed that *LuDREB49* and *LuDREB56* exhibited high expression levels in the capsules, whereas *LuDREB3* and *LuDREB36* showed significantly higher expression levels in the stems, suggesting that these *LuDREB* genes may have specialized functions in capsule or stem development. Collectively, this study provides a comprehensive overview of *LuDREB* genes, offering valuable insights into their roles in flax growth, development, and stress responses.

## 1. Introduction

Drastic changes in the global climate pose substantial challenges to plant growth and development by aggravating abiotic stresses such as extreme temperatures, drought, and salinization, leading to substantial agricultural losses [1,2]. Modifying the expression patterns of stress-responsive genes under these environmental stresses is critical to ensuring plant survival in adverse conditions. These genes are classified into those encoding stress tolerance proteins and those encoding regulatory proteins [3]. Transcription factors (TFs) play a key role in regulating the expression of stress-response genes by specifically binding to *cis*-acting elements in their promoter regions [4,5]. Several TF families in plants, including WRKY, MYB, and AP2/ERF (APETALA2/ethylene-responsive factor), are induced under abiotic stress [6,7,8]. Based on the number of domains and specific binding sequences, the AP2/ERF family is further divided into four major groups: the AP2 subfamily, the ERF subfamily, the DREB (dehydration-responsive element-binding protein) subfamily, and the RAV subfamily [9,10].

*DREB* genes play a central role in response to various abiotic stresses, including drought, high salinity, cold, and heat [11,12,13]. Members of the *DREB* gene family contain a single AP2 domain that interacts specifically with the core motif (A/GCCGAC) of the dehydration-responsive element (DRE) and binds to the C-repeat (CRT) *cis*-acting element, regulating the expression of downstream genes and enhancing plant tolerance to abiotic stresses (Figure 1) [14,15]. For instance, 5 out of 30 *DREB* genes in mung bean showed significant upregulation under drought conditions [16]. *AtDREB1A* and *AtDREB1B* from *Arabidopsis thaliana* positively affect drought tolerance in transgenic *Salvia miltiorrhiza* [17,18]. Similarly, the overexpression of the soybean *DREB1* gene in wheat enhances drought tolerance in field conditions [19]. In addition to drought stress, *DREB* genes are also activated in response to low temperatures, heat, and high salinity conditions [20,21,22,23,24,25,26,27]. The overexpression of *TaDREB3* in wheat improves plant tolerance to heat, dehydration, and salt stress [28], and the expression of several *DREB* genes in barley is significantly upregulated under drought and salt stimulations [29]. In *Arabidopsis*, *DREB2A* and *DREB2B* exhibit positive responses to dehydration and high salinity [30]. Furthermore, *SsDREB* genes in *Saccharum spontaneum* have been linked not only to stress responses but also to development and photosynthesis [31]. In rice, the expression of *OsDREB1C* is induced by both light and low-nitrogen conditions, with its overexpression enhancing grain yield and nitrogen use efficiency and promoting early flowering [32]. Collectively, *DREB* genes serve as key regulators in plant adaptation to abiotic stresses.

Flax (*Linum usitatissimum* L.) belongs to the genus *Linum* within the Linaceae family. It is a self-pollinating, annual, diploid herbaceous dicotyledonous plant (2n = 2x = 30) [33]. Flax serves as a vital oilseed and cash crop in China, exhibiting strong resistance to drought, cold, and nutrient-deficient conditions [34]. Flax has been extensively used in industries such as oil production, textiles, printing, tanning, pharmaceuticals, and food, highlighting its substantial economic value [35]. A bioinformatics analysis of the flax genome identified 43,668 protein-coding genes [36]. Numerous gene families in flax have been identified and characterized to date. For example, the WRKY transcription factor family in flax has been comprehensively identified and analyzed at the whole-genome level, along with its expression patterns [37]. Additionally, 167 R2R3-, 7 3R-, and 1 4R-MYB transcription factors have been identified in flax. Expression analysis indicates that eight R2R3 MYB genes may be involved in lignin biosynthesis [38]. In addition, 50 *LuLEA* genes were identified, with expression profiles indicating that most of these genes play a role in seed development [39].

Although substantial research has been conducted on the flax genome, the genetic basis underlying its crucial agronomic traits and adaptations to environmental stresses remains poorly understood, and *DREB* genes within the flax genome have yet to be studied. Given the critical role of *DREB* genes in mediating responses to environmental stresses across various species, studying the *DREB* gene family in flax (*LuDREB*) holds importance. Through a comprehensive genome-wide analysis, this study identified and characterized the *LuDREB* gene family, focusing on the chromosomal location, gene structure, conserved motifs, *cis*-acting elements, evolutionary relationships, and expression patterns across different tissues and under drought stress. The findings of this study provide a detailed understanding of the *LuDREB* gene family and establish a foundation for elucidating the molecular mechanisms underlying drought tolerance in flax.

## 2. Results

### 2.1. Identification of DREB Genes in Flax

Orthologous genes of *DREB* from *Arabidopsis* were screened in the genomes of Longya-10 and pale flax [40]. A total of 59 and 57 *DREB* members, designated *LuDREB1* to *LuDREB59*, were identified in Longya-10 and pale flax, respectively (Appendix A). Further analysis revealed that the protein lengths of DREB members ranged from 123 (LuDREB17) to 973 (LuDREB47) amino acids (aa) in Longya-10 and from 121 to 959 aa in pale flax. The MW ranged from 13.16 (LuDREB17) to 109.95 kDa (LuDREB47) in Longya-10 and from 13.17 to 108.63 kDa in pale flax. The pI ranged from 4.23 to 9.36 in Longya-10 and from 4.71 to 9.52 in pale flax, with most DREB members classified as acidic proteins. Moreover, the GRAVY values for all DREB proteins were less than zero, indicating their hydrophilic nature. A subcellular localization analysis indicated that 23 and 20 DREB members were located in both the cytoplasm and nucleus in Longya-10 and pale flax, respectively, whereas 35 members were exclusively located in the nucleus in both genomes. Only one DREB member in Longya-10 and two in pale flax were found to be located solely in the cytoplasm. The DREB protein sequences are listed in Appendix A.

### 2.2. Phylogenetic Analysis of LuDREB Proteins

To examine the evolutionary relationships of flax *DREB* genes, a phylogenetic tree was constructed using *DREB* sequences from the breeding cultivar Longya-10 and its wild ancestor, pale flax (Appendix A). The analysis revealed that most *LuDREB* genes had orthologs in pale flax, suggesting a high degree of conservation of *DREB* genes during flax domestication.

To further investigate the evolution of *DREB* genes across diverse species, a dataset comprising 290 *DREB* sequences was compiled, including 59, 20, 16, 56, 73, and 66 sequences from flax, maize, rice, *Arabidopsis*, soybean, and potato, respectively. A comprehensive phylogenetic tree was constructed based on this dataset (Figure 2). The *DREB* sequences were classified into six subgroups (Groups 1 to 6) following the *Arabidopsis* classification system [9]. The *LuDREB* genes were unevenly distributed across these subgroups, with Group 4 containing the highest number (28 members) and Group 3 containing the fewest (2 members). Groups 1, 2, 5, and 6 included 3, 5, 9, and 12 *LuDREB* members, respectively. Groups 1, 2, and 3 contained *DREB* genes from both monocotyledonous and dicotyledonous plants, whereas Groups 4, 5, and 6 exclusively comprised *DREB* genes from dicotyledonous plants. Notably, the *Arabidopsis* gene *AT2G40220.1* (*ABI4*) exhibited high homology with *DREB* genes from all five other species in Group 3, indicating the strong conservation of *ABI4* across species and highlighting its significant and conserved role in plants.

### 2.3. Sequence Analysis of Flax DREBs

To evaluate genetic relationships among LuDREB proteins, a phylogenetic tree was constructed based on the amino acid sequences of all LuDREB proteins, which were subsequently classified into six groups (Groups 1 to 6), with most members existing in pairs (Figure 3a). The majority of LuDREB proteins displayed high sequence similarity to another member, forming pairs. The gene structure analysis revealed significant variation in the length of *DREB* genes. In Longya-10, the lengths ranged from 372 (*LuDREB17*) to 5636 bp (*LuDREB47*), while, in pale flax, the lengths varied from 392 bp to 10,637 bp (Figure 3b, Appendix A). The exon–intron organization of all *DREB* genes in flax was also examined (Figure 3b). The analysis indicated that most *LuDREB* family members (78%) contained only one exon. Among the remaining genes, 13 had more than one exon: 5 genes contained two exons (*LuDREB11*, *LuDREB14*, *LuDREB36*, *LuDREB51*, and *LuDREB56*), 2 contained four exons (*LuDREB49* and *LuDREB57*), 2 contained seven exons (*LuDREB9* and *LuDREB52*), and 3 had twelve exons (*LuDREB25*, *LuDREB34*, and *LuDREB47*). Notably, *LuDREB4* had the highest number of exons (13). In pale flax, 43 out of 57 *DREB* genes contained only one exon, whereas the remaining genes possessed 2–13 exons.

The MEME program was used to identify conserved motifs within LuDREB proteins (Figure 3c). A total of 80 conserved motifs were identified, ranging in length from 8 to 50 aa. The motif composition varied across subgroups. Motifs 1, 2, and 3 were present in the majority of LuDREB proteins, with a few exceptions, whereas the remaining motifs were specific to certain LuDREB members. Motifs 1 and 2 were found in all members of the five subgroups except Group 4, where 11 and 16 out of the 28 LuDREB members lacked Motifs 1 and 2, respectively. Motif 3 was present in all members of Groups 1, 2, and 6, as well as most members of Groups 4 and 5, but it was absent in Group 3. In addition, Motif 4 was found in Groups 1, 4, and 5, whereas Motif 5 was restricted to Groups 1 and 4. Several motifs were unique to specific subgroups or individual members. For instance, Group 4 contained up to 36 subgroup-specific motifs, whereas Motifs 16, 37, and 55 were exclusive to Groups 6, 5, and 2, respectively. Significant variation in motif composition was observed within Group 4, whereas LuDREB members in the other five subgroups displayed a more consistent motif pattern. For example, all members of Group 1 contained Motifs 1–5, whereas members of Group 3 shared Motifs 1, 2, 27, 30, 31, 47, and 66 (Appendix A).

### 2.4. Chromosomal Location and Synteny Analysis

The genomic chromosomal location analysis of the *LuDREB* genes revealed an uneven distribution of 56 *LuDREB* genes across all 15 chromosomes (Figure 4). However, *LuDREB57*, *LuDREB58*, and *LuDREB59* could not be mapped to any chromosome. Chromosome 10 contained the highest number of *LuDREB* genes (six genes), whereas chromosome 8 harbored only two genes. Chromosomes 1, 2, 4, 5, 6, 11, and 14 each contained three genes, chromosomes 3, 7, and 9 each harbored four genes, and chromosomes 12, 13, and 15 contained five genes each.

An analysis of duplication events was conducted to elucidate the mechanisms underlying the expansion of the *LuDREB* gene family during evolution (Figure 4 and Table 1). The results showed that one gene pair, *LuDREB42/LuDREB43*, located on chromosome 12, underwent tandem duplication (TD). Furthermore, segmental duplication (SD) was observed in 30 gene pairs, involving a total of 51 *LuDREB* genes, which accounted for 86.4% of the entire *LuDREB* family. To assess the selective pressure acting on duplicated *LuDREB* gene pairs, the K_a_ and K_s_ substitution rates and K_a_/K_s_ ratios were calculated. K_a_/K_s_ ratios of 1, greater than 1, and less than 1 indicate neutral, positive, and purifying selection, respectively [41]. The K_a_/K_s_ ratios for all duplicated gene pairs were less than 0.5, indicating that purifying selection shaped the evolution of *LuDREB* genes. The distribution of K_s_ values can be utilized to identify potential genome duplication events and estimate the timing of such events, as synonymous mutations are generally considered neutral and are not subject to natural selection [42]. By comparing the K_s_ value distributions of duplicated gene pairs within a genome, whole-genome duplication (WGD) events can be detected, and the frequency and distribution of these WGD events throughout evolutionary history can be inferred. This is because WGD typically results in the simultaneous duplication of a large number of genes, leading to a concentration of K_s_ values within a relatively narrow range [36,43]. Our study shows that 80% of the collinear genes have K_s_ values ranging from 0.07 to 0.31, suggesting that the expansion of *LuDREB* genes likely originated from a recent WGD event in flax (K_s_ = 0.13). The duplication events of these genes were estimated to have occurred between 6 and 25 Mya, whereas the divergence time of other genes was estimated to be between 78 and 183 Mya.

To further explore the phylogenetic and evolutionary relationships of *DREB* genes across species, a collinearity analysis was conducted using three dicots (*Arabidopsis*, potato, and soybean) and two monocots (maize and rice) (Figure 5 and Appendix A). The analysis revealed that 39, 33, 44, 9, and 3 out of 56 *LuDREB* genes had orthologs in *Arabidopsis*, potato, soybean, maize, and rice, respectively, with 58, 50, 121, 13, and 4 collinear gene pairs identified between flax and these species. These findings indicate that more than half of the *LuDREB* genes had orthologs in the three dicot plants, and the collinearity between flax and dicots was greater than that observed with monocots. In addition, homologous *DREB* genes exhibited one-to-many and many-to-one collinear relationships. For instance, *LuDREB14*, *LuDREB35*, and *LuDREB36* had up to five collinear gene pairs in soybean, whereas *LuDREB36* had four pairs in potato, demonstrating a high degree of collinearity. Furthermore, *LuDREB30* and *LuDREB40* had orthologs in all five species, suggesting that these genes originated from a common ancestor and their functions are conserved in both dicots and monocots. Notably, 20 *LuDREB* genes had homologs in all dicots but lacked orthologs in monocots, indicating their potential significance in the evolutionary divergence between dicots and monocots.

### 2.5. Cis-Acting Elements on LuDREB Promoters

To investigate the transcriptional regulation mechanisms and potential functions of *LuDREB* genes, the 2000-bp upstream sequences of each gene were extracted to predict *cis*-acting elements in their promoters (Figure 6 and Appendix A). The analysis identified seven types of stress-responsive elements on *LuDREB* promoters, including MBS (drought inducibility), CCAAT-box (heat shock), LTR (low temperature response), GC-motif and ARE (anoxia response), TC-rich repeats (defense and stress responses), WUN-motif (wound response), and AT-rich sequences (other abiotic stress responses). More than half of the *LuDREB* genes contained elements associated with drought (32 genes), low temperature (38 genes), and anoxia responsiveness (55 genes). The total number of these elements was 43, 57, and 157, respectively. Among the 55 genes, *LuDREB58* had the highest number of anoxia-responsive elements. In addition, elements related to abscisic acid (190), methyl jasmonate (MeJA) (119), gibberellin (50), auxin (47), and salicylic acid (41) were abundant in *LuDREB* promoters, although their distribution was uneven. The number of phytohormone-related elements per gene ranged from 1 (*LuDREB36*) to 19 (*LuDREB6*). All *LuDREB* genes contained at least one phytohormone-responsive element. Specifically, *LuDREB10*, *LuDREB27*, *LuDREB31*, *LuDREB34*, *LuDREB38*, *LuDREB56*, and *LuDREB59* contained all five phytohormone-responsive elements, whereas *LuDREB36* contained only one type. *LuDREB6*, *LuDREB18*, and *LuDREB51* had up to 9, 7, and 4 elements associated with abscisic acid, MeJA, and gibberellin, respectively.

### 2.6. Expression Analysis of LuDREB Genes in Different Tissues of Flax Under Drought Stress

To investigate the potential functions of *LuDREB* genes in flax development, their expression profiles were analyzed across different cultivars (Longya-10 and Heiya-14) and tissues (stem and capsule), using previously reported transcriptome data (Figure 7) [36]. The analysis revealed the presence of the transcripts of 56 *LuDREB* genes in at least one organ in one of the two cultivars. Among these, 43 genes were expressed in both the stems and capsules of both cultivars. However, the transcripts of *LuDREB17*, *LuDREB18*, and *LuDREB38* were not detected. Notably, *LuDREB49* and *LuDREB56* showed high expression levels in the capsules of both cultivars, with *LuDREB56* exhibiting particularly strong expression. By contrast, five genes (*LuDREB3*, *LuDREB10*, *LuDREB23*, *LuDREB36*, and *LuDREB53*), especially *LuDREB3* and *LuDREB36*, demonstrated significantly higher expression levels in the stems of both cultivars. The expression levels of other *LuDREB* genes were relatively low across both cultivars.

To evaluate the potential functions of *LuDREB* genes in response to drought stress, the expression levels of genes containing MBS (drought-related *cis*-acting elements) in their promoters were analyzed under drought stress simulated by PEG treatment (Figure 8). The results showed that all analyzed *LuDREB* genes responded to drought stress to varying degrees, with each displaying distinct spatiotemporal patterns. Most *LuDREB* genes were upregulated at least at one time point in either the roots or leaves after exposure. Notably, the expression levels of *LuDREB23* exhibited an upward regulation trend at all time points in both the roots and leaves following treatment. *LuDREB11/51*, *LuDREB13*, *LuDREB17*, *LuDREB22*, *LuDREB33*, *LuDREB35*, and *LuDREB50* exhibited an upward regulation trend in leaves across all time points. Among them, *LuDREB11/51* and *LuDREB13* reached their peak expression levels at 72 h, indicating their potential involvement in late-phase signaling pathways under drought stress. In contrast, the expression levels of *LuDREB1* showed a consistent downward trend at all time points in leaves after treatment, potentially reflecting its negative regulatory role under drought conditions. Based on these findings, it can be inferred that different *LuDREB* genes regulate drought responses through distinct mechanisms. We also observed distinct expression patterns of *LuDREB* genes between roots and leaves, which may reflect the complexity of drought response mechanisms and suggest potential functional divergence of *LuDREB* genes in different tissues. For instance, *LuDREB10*, *LuDREB2/20*, and *LuDREB27* had lower expression levels in leaves than in roots at all time points after PEG treatment, whereas the expression levels of *LuDREB35* and *LuDREB50* were significantly higher in leaves than in roots.

## 3. Discussion

Crop yield and quality are significantly affected by adverse environmental conditions [9]. However, plants have evolved various adaptive mechanisms to resist environmental stresses. Among these, DREB TFs play pivotal roles in mediating abiotic stress responses through both direct and indirect pathways [44,45,46]. The *DREB* gene family has been identified in numerous plant species. For instance, 56 *DREB* genes have been identified in *Arabidopsis* genome [9], whereas the genomes of soybean and potato contain 73 and 66 *DREB* members, respectively [47,48]. In the present study, 59 and 57 DREB members were identified in the genomes of flax: specifically, in the breeding variety Longya-10 and its ancestor, pale flax. These findings underscore the widespread distribution of the *DREB* gene family across species. However, variations in the number of DREB members among species, likely driven by gene expansion and deletion events, highlight evolutionary diversity and reflect the specific environmental adaptations required by different plants. Furthermore, the similar number of DREB genes in the two flax genomes suggests that the DREB gene family remained relatively conserved during flax domestication.

To better understand the potential functions of proteins, analyzing their physicochemical properties is essential [49]. DREB proteins in flax vary significantly in terms of length, gene structure, MW, and pI, which suggest their diverse roles in plant development and defense responses. Moreover, most flax *DREB* genes contained few or no introns. Previous studies have demonstrated that genes with minimal or no introns often exhibit enhanced expression levels in plants [50,51]. In addition, compact gene structures enhance the timely response of plants to various abiotic stresses [52]. Notably, *LuDREB23* has a gene structure comprising only one exon, and this compact structure may facilitate its rapid transcription during the early stages of stress. The predicted subcellular localization of flax DREB proteins indicated that they were either distributed in both the nucleus and cytoplasm or restricted to one of these locations. These findings are consistent with those of previous studies on DREB proteins [47,53]. Conserved motifs are crucial for predicting protein functions [7]. Among the 80 motifs identified in LuDREB proteins, Motifs 1, 2, and 3 were present in most members, suggesting that these motifs are conserved within the LuDREB family. Notably, Motif 1, which forms part of the AP2 domain, aligns with previous research findings [31,54]. By contrast, other motifs found exclusively in individual LuDREB members may contribute to functional diversity.

In this study, three phylogenetic trees were constructed to investigate evolutionary relationships among *LuDREB* members, classifying them into six subgroups with an uneven distribution. Notably, Groups 1, 2, and 3 included *DREB* genes from both monocotyledonous and dicotyledonous plants, suggesting that the functions of these members are conserved between monocots and dicots. By contrast, Groups 4, 5, and 6 exclusively comprised *DREB* genes from dicotyledonous plants, indicating their specialized roles in dicots. Moreover, closely related members shared similar gene structures and motif distributions. The analysis revealed that *LuDREB* members in Groups 1 and 2 exhibited high homology with *Arabidopsis* genes *DREB1A*, *DREB1B*, *DREB1C*, *DREB2A*, and *DREB2B*, which are induced by low temperatures, drought, high salinity, or heat stress [55]. This finding suggests that these *LuDREB* genes also participate in stress responses. Furthermore, *LuDREB45* and *LuDREB55* were closely related to *AtABI4* and *ZmABI4*, which are involved in abscisic acid (ABA) signaling, seed maturation, lateral root formation, and sugar signaling [56]. Combined with the presence of ABA-responsive elements in their promoters, *LuDREB45* and *LuDREB55* are likely involved in ABA signaling.

WGD or polyploidization is a key driver of evolution, generating novel traits and transcriptional regulatory factors that affect the expression patterns of downstream genes [57]. Numerous studies have demonstrated that SD, WGD, and TD contribute significantly to gene expansion and functional diversification within multigene families [58,59]. While SD often leads to functional redundancy, TD can generate novel functions, enabling species to adapt to rapidly changing environments [60]. In this study, we observed that two *LuDREB* genes underwent TD, whereas SD was observed in 51 genes (86.4%, forming 30 gene pairs). This suggests that functional redundancy within the flax *DREB* family may serve as a buffering mechanism to cope with fluctuating environments [60]. This finding is consistent with the expansion pattern of *DREB* genes observed in soybean [19]. K_s_ analysis further indicated that the expansion of the *LuDREB* family was primarily driven by a recent WGD event [36]. Moreover, all duplicated *LuDREB* gene pairs exhibited K_a_/K_s_ ratios of less than 1, suggesting that *LuDREB* genes have undergone purifying selection throughout their evolution. Collinearity analysis between flax and five other species (three dicots and two monocots) revealed that most *LuDREB* genes had orthologs in these species, highlighting the evolutionary importance and relative conservation of *DREB* genes. Furthermore, the number of collinear genes between flax and dicotyledonous plants was significantly higher than that observed between flax and monocotyledonous plants. This finding suggests that *DREB* genes have undergone widespread duplication events following the divergence of dicots and monocots, potentially contributing to the unique characteristics of these two plant groups.

*DREB* genes contain numerous hormone- and stress-responsive elements, allowing them to be recognized by upstream signaling molecules, thereby regulating their own expression as well as that of downstream genes. This regulation enables plants to withstand adverse environmental conditions [30,61]. Extensive research has demonstrated that DREB TFs enhance plant resistance to abiotic stresses. For example, transgenic plants overexpressing *OsDREB1A*, *OsDREB1B*, and *OsDREB1F* exhibit increased tolerance to drought stress [62,63,64]. Similarly, Chen et al. (2008) reported that the overexpression of *OsDREB1G* significantly improves drought tolerance in rice [65]. The heterologous expression of *BrDREB2B* also enhanced salt, heat, and drought tolerance in *Arabidopsis* [15]. In addition, *DREB* genes regulate the expression of stress-responsive genes through both ABA-dependent and ABA-independent pathways [66]. The response of plants to abiotic stresses typically involves complex regulatory networks, where the synergistic interactions between DREB TFs and other TF families (e.g., MYB, WRKY, and NAC), as well as hormone signaling pathways (e.g., ABA and JA), play crucial roles [67]. In this study, an abundance of hormone- and stress-responsive elements was identified in the promoters of *LuDREB* genes, indicating their potential involvement in stress responses via crosstalk with the ABA or JA signaling pathways. For instance, the presence of ABRE elements suggests that *LuDREB* genes may be regulated in an ABA-dependent manner, whereas MeJA-responsive elements indicate that jasmonic acid signaling may indirectly regulate downstream defense genes by activating *LuDREB* expression [44]. Furthermore, the synergistic interactions between DREB and other TFs, such as WRKY or MYB, have been extensively documented in various plants. For example, in *Arabidopsis*, DREB2A and WRKY18 enhance drought responses through co-binding to the promoter regions of downstream genes [27]. Similarly, the WRKY family members identified in flax may form complex regulatory modules with LuDREBs via analogous mechanisms [37]. Moreover, the synergistic interaction between DREB and NAC TFs also plays a pivotal role in drought signal transduction. For instance, the co-expression of *OsDREB1F* and *OsNAC6* in rice significantly enhances drought tolerance [65]. In the flax genome, multiple NAC family members have been characterized [35], and their potential interactions with LuDREBs merit further investigation. Such multi-factor synergistic interactions may amplify signal transduction pathways or integrate multiple stress signals, thereby enhancing plant adaptability to complex environmental stresses [31].

To further explore their involvement in drought responses, the expression profiles of *LuDREB* genes containing MBS elements were analyzed under drought stress. The findings indicated that all analyzed *LuDREB* genes exhibited varying degrees of response to PEG treatment. Previous studies have reported similar findings in other species. For instance, four *SsDREB* genes (*SsDREB1F*, *SsDREB1L*, *SsDREB2D*, and *SsDREB2F*) were induced by drought stress in three sugarcane varieties [31], whereas, in sugar beet, nine *BvDREB* genes showed significant upregulation under drought stress [68]. In *Lotus japonicus*, three *DREB* A-2 subgroup genes were rapidly induced by drought stress [54]. These observations suggest the presence of synergistic effects or network regulatory mechanisms that enhance drought resistance in plants [69]. We also observed that the expression patterns of *LuDREB* genes varied significantly between leaves and roots. Previous studies have demonstrated that the upregulation of *DREB* gene expression in plant roots is mainly associated with promoting root growth and enhancing water uptake [70], whereas their expression in leaves is linked to stomatal regulation and reduced transpiration [71,72,73]. This implies that the upregulation of *LuDREB10*, *LuDREB2/20*, and *LuDREB27* in roots may reflect their roles in modulating root growth or improving water uptake efficiency. The upregulation of *LuDREB35* and *LuDREB50* in leaves could be related to reduced transpiration and stomatal closure, which are essential for water conservation in aerial tissues. Additionally, the concurrent upregulation of *LuDREB23* in both roots and leaves suggests its potential involvement in general drought response pathways, such as osmotic adjustment and stress signaling. These findings underscore the complexity of drought response mechanisms and highlight the potential functional divergence of *LuDREB* genes across different tissues. Notably, most *LuDREB* genes containing MBS elements also harbored ABRE elements, suggesting that *LuDREB* genes may mediate drought responses through the ABA signaling pathway, as observed in other plant species [74]. In this study, *LuDREB23*, *LuDREB11*, and *LuDREB51* were consistently upregulated under drought stress, and their promoter regions were enriched in MBS and ABRE, indicating that they may regulate downstream genes via both ABA-dependent and ABA-independent pathways [66]. In addition, different *LuDREB* genes exhibited distinct expression patterns in response to drought stress, indicating that they may function through multiple regulatory mechanisms. This study also generated a heatmap using transcriptome data to preliminarily predict the roles of *LuDREB* genes in organ development. The results indicated that *LuDREB56* may be involved in capsule development, whereas *LuDREB3* and *LuDREB36* may play roles in stem development. However, further functional validation experiments are needed to clarify the specific roles of individual *LuDREB* genes.

Although this study systematically identified and analyzed the *LuDREB* family and elucidated its drought response patterns, certain limitations are acknowledged. First, the qRT-PCR analysis was restricted to genes containing MBS elements, and it may have overlooked key members of other regulatory pathways. Second, the functional validation of genes relied solely on expression profile data, lacking direct experimental evidence from transgenic and gene-editing experiments. In future research, we plan to utilize CRISPR-Cas9 to knockout specific genes or heterologously overexpress them in *Arabidopsis* to clarify their roles in conferring drought tolerance. Additionally, the long-term domestication of flax may have resulted in unique *DREB* allelic variations, and integrating population genetic analyses (e.g., GWAS) will help to determine their breeding potential.

## 4. Materials and Methods

### 4.1. Plant Growth and Drought Treatment

The flax variety Longya-10 was used in this study. To examine the expression patterns of *LuDREB* genes under drought stress, Longya-10 seeds were germinated in Petri dishes maintained in a climate chamber at 25 °C with a 16:8 light/dark photoperiod. After 14 days of growth, uniform seedlings were selected and transferred to 1/2 Murashige and Skoog (MS) liquid medium for the additional 4 days of cultivation. Then, drought stress was induced by replacing the medium with a fresh medium supplemented with 20% (*w*/*v*) polyethylene glycol (PEG). All plants subjected to the PEG treatment originated from the same sowing, and the experiment was conducted in three biological replicates. The leaves and roots were collected at 0, 3, 6, 9, 12, 24, 48, and 72 h during the PEG treatment and at 48 h after recovery in the 1/2 MS liquid medium without PEG supplementation [54,75]. The collected samples were immediately frozen in liquid nitrogen and stored at −80 °C for subsequent quantitative real-time reverse transcription polymerase chain reaction (qRT-PCR) analysis.

### 4.2. Identification of DREB Genes

The genome sequences of *Linum usitatissimum* (Longya-10) and *Linum bienne* (pale flax) utilized in this study were obtained from our laboratory and have been deposited in the DDBJ/ENA/GenBank under the accession numbers QMEI00000000 and QMEG00000000 [36]. Their annotation files were downloaded from Figshare (https://figshare.com/ (accessed on 18 December 2024)). The *Arabidopsis* DREB protein sequences were retrieved from the *Arabidopsis* Information Resource (TAIR, https://www.arabidopsis.org/ (accessed on 18 December 2024)), and DREB protein sequences for *Zea mays*, *Solanum tuberosum*, *Oryza sativa*, and *Glycine max* were obtained from previous studies [10,47,48,76,77,78]. OrthoFinder 2.5.5 was used to identify orthologs of the *Arabidopsis DREB* gene family in Longya-10 and pale flax [40]. The molecular weight (MW), isoelectric point (pI), grand average of hydropathicity (GRAVY), and protein length (amino acids) of all LuDREB proteins were calculated using the ExPASy ProParam tool (http://web.expasy.org/protparam/ (accessed on 20 December 2024)) [79]. The subcellular localization of the LuDREB proteins was predicted using Plant-mPloc (http://www.csbio.sjtu.edu.cn/bioinf/plant-multi/ (accessed on 20 December 2024)) [80]. Conserved motifs within LuDREB proteins were identified and analyzed using the MEME suite (https://meme-suite.org/meme/tools/meme (accessed on 21 December 2024)), with default parameter settings [81].

### 4.3. Chromosomal Mapping and Collinearity Analysis

The chromosomal location information for the *LuDREB* genes was retrieved from the annotation files of the Longya-10 genome, and visualization was performed using the Map Gene 2 Chromosome (MG2C, http://mg2c.iask.in/mg2c_v2.0/ (accessed on 24 December 2024)) tool. The MCScanX package was used to analyze *LuDREB* gene duplication events with default parameters, whereas TBtools software (v. 2.146) was used to analyze and visualize collinearity among *DREB* genes across different species [82,83]. The nonsynonymous (K_a_) and synonymous (K_s_) substitution rates per site for the duplicated gene pairs were calculated using TBtools software (v. 2.146), and the K_a_/K_s_ ratio was determined to assess the mode and intensity of selective pressure. For each gene pair, the divergence time (million years ago, Mya) was calculated using the following formula [84]:Mya = K_s_/(2 × 6.1 × 10^−9^) × 10^−6^.

### 4.4. Gene Structure and Promoter Analysis

A 2000 bp sequence upstream of the transcription start site was analyzed to identify *cis*-acting elements within the *LuDREB* gene family. *Cis*-acting regulatory elements were predicted using the PlantCARE website (http://bioinformatics.psb.ugent.be/webtools/plantcare/html/ (accessed on 26 December 2024)), and the results were visualized using TBtools software (v2.146) [85]. The exon–intron organization of the *LuDREB* gene family was determined and analyzed using the Gene Structure Display Server (GSDS, http://gsds.gao-lab.org/ (accessed on 29 December 2024)) [86].

### 4.5. Phylogenetic Analysis

DREB protein sequences from rice, maize, *Arabidopsis*, soybean, and potato were used to construct the phylogenetic tree [10,47,48,76,77,78]. The sequence alignment of DREB proteins was performed using ClustalW in MEGA (v. 11.0), with the default parameters. The phylogenetic tree was subsequently constructed based on the sequence alignment using the neighbor-joining method, employing the Poisson model and pairwise deletion, with 1000 bootstrap replicates [87]. Finally, R version 4.1.3 was employed to visualize the phylogenetic tree.

### 4.6. Gene Expression Analysis

The transcriptome data generated in our lab (PRJNA505721) were used to analyze the expression patterns of *LuDREB* genes across different genetic backgrounds and tissues [36]. The expression profiles of *LuDREB* genes under the PEG treatment were further examined using qRT-PCR. Total RNA was extracted using the EZgene Plant Easy Spin RNA Miniprep Kit (BIOMIGA, San Diego, CA, USA), and cDNA was synthesized using the PrimeScript RT Reagent Kit with gDNA Eraser (Takara, San Jose, CA, USA). Specific primers for qRT-PCR were designed using Primer Premier 5.0 (PREMIER Biosoft International, Palo Alto, CA, USA) (Appendix A). The qRT-PCR experiment was conducted using the Eco Real-Time PCR System (Illumine, San Diego, CA, USA) following the protocol described by Qi et al. (2023) [75], using *GAPDH* as the reference gene [88]. The relative expression levels of *LuDREB* genes were calculated using the 2^−∆∆Ct^ method, and the data were subsequently subjected to logarithmic transformation (log10), with the leaves collected at 0 h serving as the control [89]. Because of the high sequence similarity among certain gene pairs (*LuDREB16/58*, *LuDREB17/40*, *LuDREB18/41*, *LuDREB46/54*, *LuDREB12/33*, and *LuDREB15/38*), it was challenging to design specific primers for individual genes. Therefore, a single primer pair was designed for each gene pair.

## 5. Conclusions

This study identified 59 *LuDREB* genes in Longya-10 (a breeding variety) and analyzed their protein characteristics, gene structure, evolutionary relationships, *cis*-acting elements, and expression patterns. The findings revealed that *LuDREB* promoters contained numerous abiotic-stress- and hormone-responsive *cis*-acting elements. In addition, the expansion of *LuDREB* genes was primarily driven by WGD events. The qRT-PCR analysis further demonstrated that the expression of multiple *LuDREB* genes increased significantly under drought stress. Collectively, these results highlight the critical roles of *LuDREB* genes in the evolution and drought responses of flax.

## Figures and Tables

**Figure 1 ijms-26-03074-f001:**
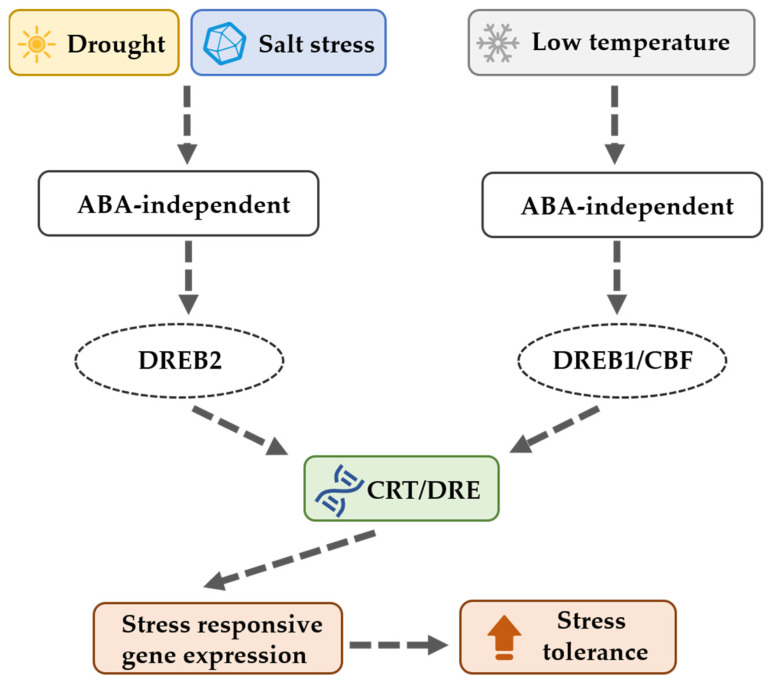
The functions of DREB transcription factors in plants.

**Figure 2 ijms-26-03074-f002:**
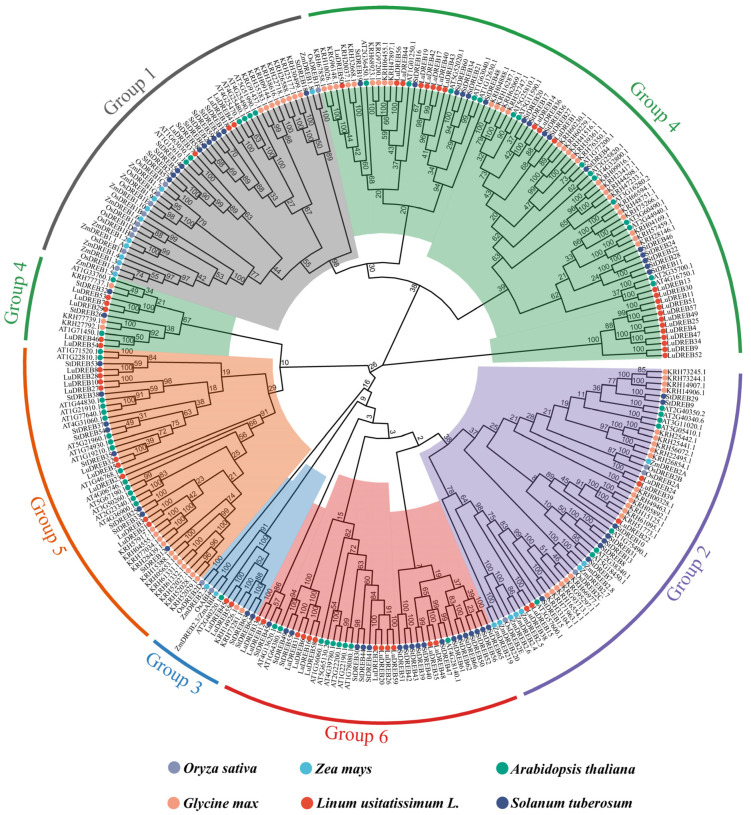
Phylogenetic tree of DREB proteins from Longya−10, rice, maize, *Arabidopsis*, soybean, and potato. The tree is divided into six clades, each represented by a different color and designated as Group 1 to Group 6. Colored circles indicate DREB members from different species.

**Figure 3 ijms-26-03074-f003:**
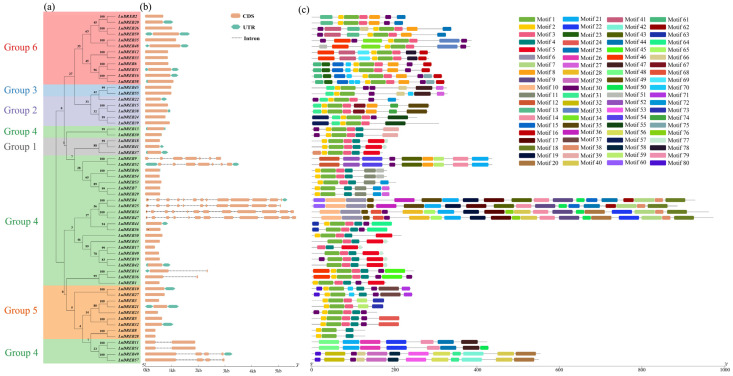
Phylogenetic relationships, gene structure, and conserved motifs of *LuDREB* genes. (**a**) Phylogenetic tree of LuDREB proteins. (**b**) Gene structure of *LuDREB* genes. (**c**) Motif composition of LuDREB proteins identified using MEME. Different colors represent distinct motifs.

**Figure 4 ijms-26-03074-f004:**
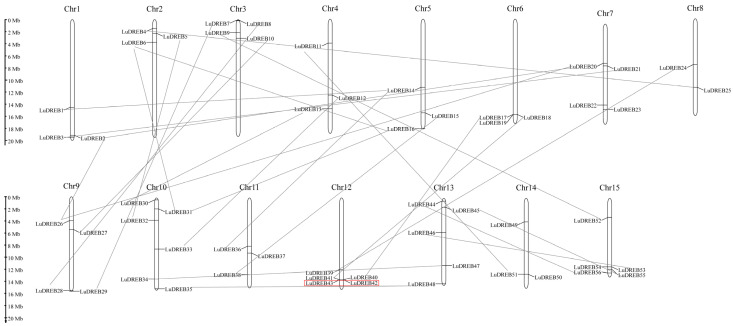
Chromosomal distribution of *LuDREB* genes. Red boxes indicate gene pairs that underwent tandem duplication, whereas gray lines represent gene pairs that underwent segmental duplication.

**Figure 5 ijms-26-03074-f005:**
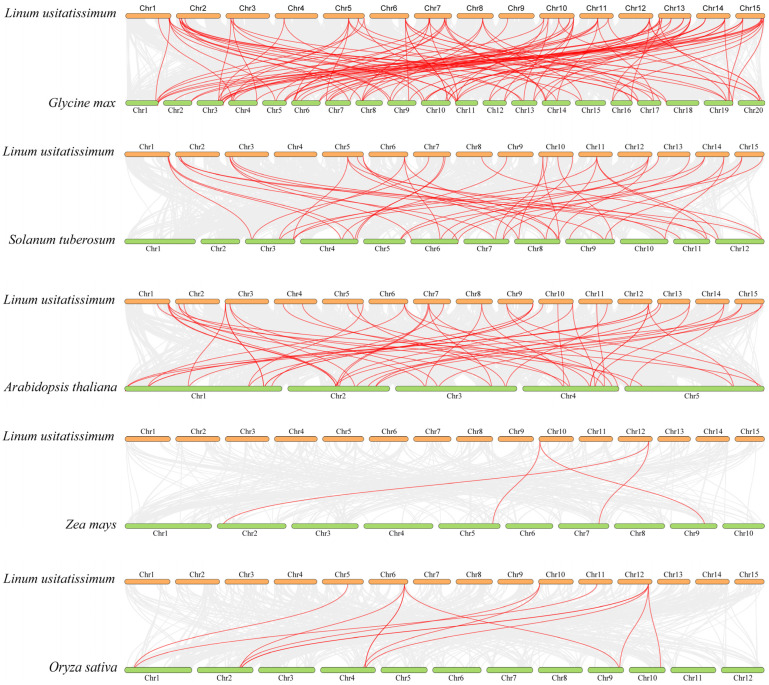
Collinearity analysis of *DREB* genes between flax and five other plant species (*Arabidopsis*, soybean, potato, maize, and rice). Gray lines indicate syntenic blocks between the flax genome and those of other species, whereas red lines represent collinear *DREB* gene pairs.

**Figure 6 ijms-26-03074-f006:**
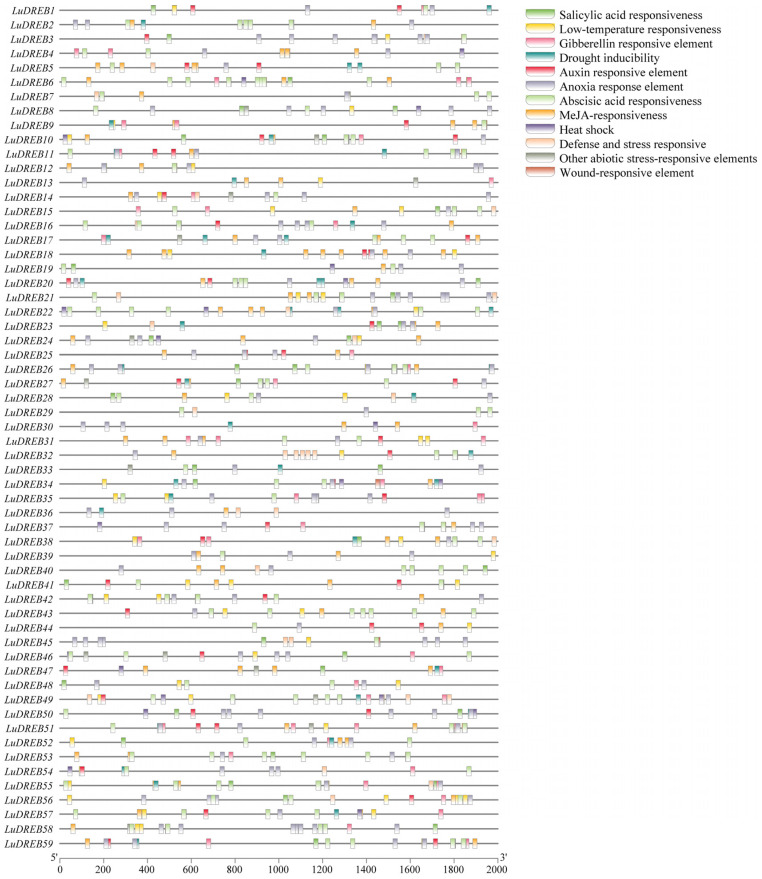
Distribution of *cis*−acting elements in the promoters of *LuDREB* genes. Different colored boxes represent distinct *cis*−acting elements.

**Figure 7 ijms-26-03074-f007:**
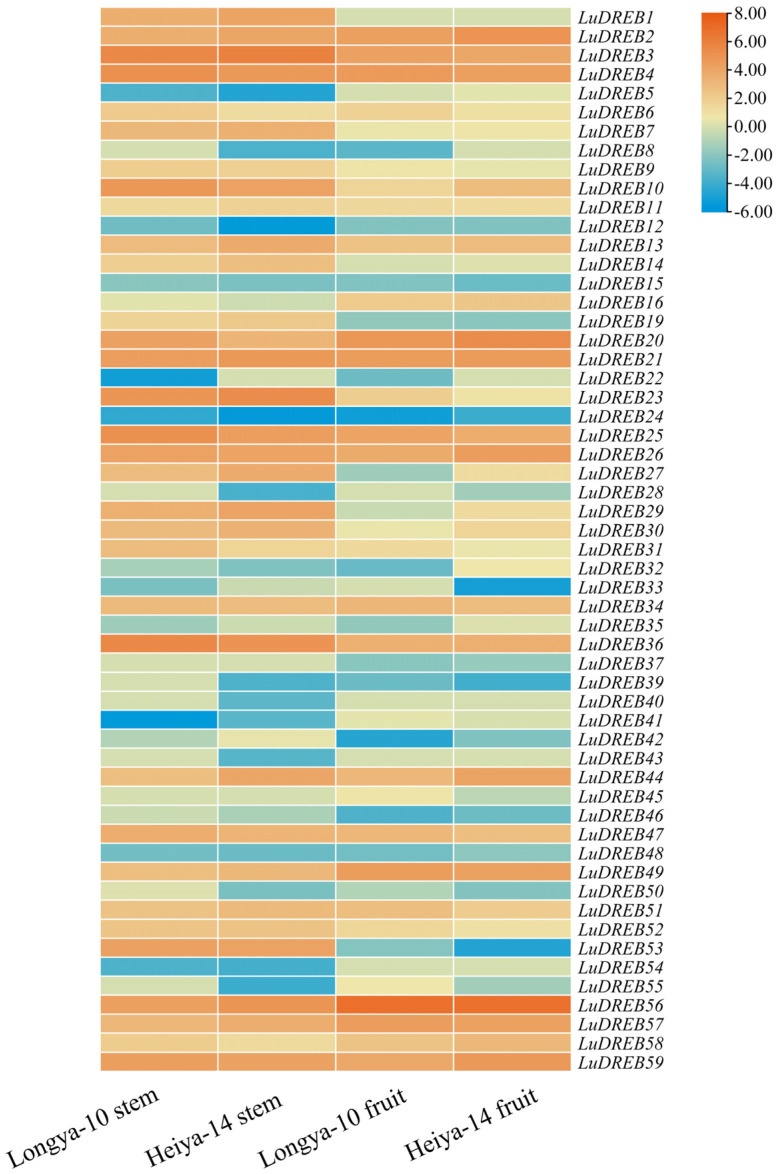
Expression analysis of *LuDREB* genes across different flax varieties and tissues based on transcriptome data (data were log2−transformed).

**Figure 8 ijms-26-03074-f008:**
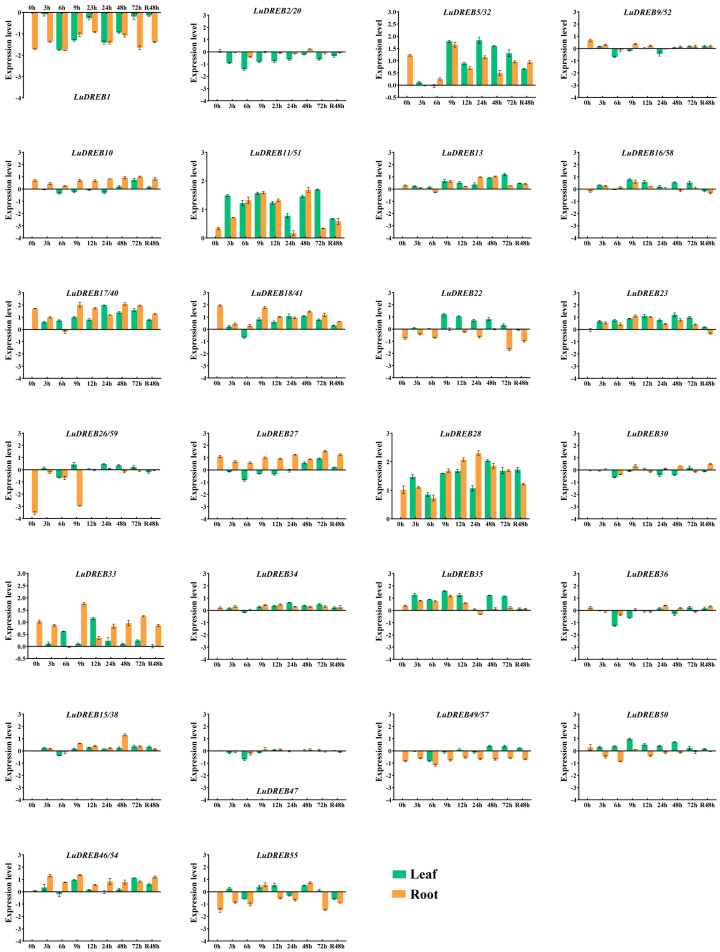
The expression levels of *LuDREB* genes in leaves and roots detected by qRT−PCR under PEG treatment. The *x*-axis represents the time points (0, 3, 6, 9, 12, 24, 48, and 72 h) after the PEG treatment and 48 h after recovery in 1/2 MS liquid medium without PEG supplementation, while the *y*-axis indicates the gene expression levels. Relative expression levels were calculated using the 2^−ΔΔCt^ method, followed by logarithmic transformation (log10). All results were derived from three biological replicates, with error bars representing ± SD (n = 3).

**Table 1 ijms-26-03074-t001:** Duplication events and divergence times of *LuDREB* genes.

Seq1	Seq2	Duplication Event	K_a_	K_s_	K_a_/K_s_	Divergence Time (Mya)
*LuDREB1*	*LuDREB14*	SD	0.2477	1.3797	0.1795	113.0911
*LuDREB2*	*LuDREB20*	SD	0.0325	0.3025	0.1074	24.7955
*LuDREB2*	*LuDREB26*	SD	0.2824	1.3108	0.2154	107.4433
*LuDREB3*	*LuDREB21*	SD	0.0104	0.0783	0.1322	6.4207
*LuDREB4*	*LuDREB25*	SD	0.0161	0.11	0.1468	9.0186
*LuDREB6*	*LuDREB16*	SD	0.1906	1.069	0.1783	87.6266
*LuDREB7*	*LuDREB29*	SD	0.0164	0.2569	0.064	21.0547
*LuDREB8*	*LuDREB28*	SD	0.037	0.1822	0.2029	14.9325
*LuDREB10*	*LuDREB27*	SD	0.0529	0.117	0.4525	9.5896
*LuDREB16*	*LuDREB58*	SD	0.0305	0.2713	0.1125	22.2348
*LuDREB20*	*LuDREB26*	SD	0.2629	2.2378	0.1175	183.4266
*LuDREB26*	*LuDREB59*	SD	0.0231	0.0747	0.3096	6.1192
*LuDREB30*	*LuDREB13*	SD	0.0714	0.2016	0.3543	16.5232
*LuDREB31*	*LuDREB6*	SD	0.0291	0.1533	0.1898	12.5668
*LuDREB31*	*LuDREB16*	SD	0.1998	1.0254	0.1949	84.0454
*LuDREB32*	*LuDREB5*	SD	0.0338	0.1559	0.2169	12.7806
*LuDREB33*	*LuDREB12*	SD	0.0297	0.1248	0.2384	10.2259
*LuDREB34*	*LuDREB47*	SD	0.0321	0.0803	0.4	6.586
*LuDREB35*	*LuDREB48*	SD	0.0628	0.2461	0.2554	20.1684
*LuDREB36*	*LuDREB14*	SD	0.0149	0.1446	0.103	11.8563
*LuDREB38*	*LuDREB15*	SD	0.0178	0.1158	0.1541	9.4944
*LuDREB39*	*LuDREB24*	SD	0.0341	0.1839	0.1857	15.0722
*LuDREB40*	*LuDREB17*	SD	0.055	0.204	0.2697	16.7172
*LuDREB41*	*LuDREB18*	SD	0.0504	0.1612	0.3129	13.2142
*LuDREB44*	*LuDREB56*	SD	0.0759	0.2569	0.2954	21.0547
*LuDREB45*	*LuDREB55*	SD	0.0911	0.3122	0.2918	25.5866
*LuDREB46*	*LuDREB53*	SD	0.3364	1.838	0.183	150.657
*LuDREB49*	*LuDREB57*	SD	0.0059	0.1234	0.0478	10.1141
*LuDREB51*	*LuDREB11*	SD	0.0593	0.1328	0.4467	10.8847
*LuDREB52*	*LuDREB9*	SD	0.0141	0.1202	0.117	9.8513
*LuDREB42*	*LuDREB43*	TD	0.2188	0.9539	0.2294	78.1916

Note: SD represents segmental duplication, and TD represents tandem duplication.

## Data Availability

All data are reported in the article and the Appendix A.

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
