# Peer review of "Genome-Wide Identification and Expression Profiling of Dehydration-Responsive Element-Binding Family Genes in Flax (Linum usitatissimum L.)"

_ijms, 2025, doi:10.3390/ijms26073074_

Round 1
Reviewer 1 Report
Comments and Suggestions for Authors
The author conducted a study on Genome-wide identification and expression profiling of dehydration-responsive element-binding family genes in flax (Linum usitatissimum L.). The research has certain significance, but the following issues exist:
Lines 32-36: Should be more specific in terms of content.
Lines 80-82: Inappropriate citations, Yuan et al. (2021), Tombuloglu (2020).
Figure 1: Lacks confidence intervals.
Lines 195-198: The statement should be more cautious. Ks values reflect gene divergence time but cannot be used alone as evidence for WGD events. More background.
Lines 202-204: Why were closely related species of flax not used?
Figure 6: The use of raw values makes it difficult to observe trends across samples. The authors should take the log for correction.
Figure 7: This figure is poorly presented. How can relative expression values be negative? Where are the error bars? qRT-PCR should be presented in its conventional format. What does "R48" represent? Why is the green signal missing at "0h" in the figure? The figure lacks a proper caption. The figure caption should be self-explanatory, allowing readers to understand it without referring to the main text.
Lines 261-273: The data lacks statistical analysis, yet phrases like "significantly upregulated" and "significantly downregulated" are used. The explanation of the figures is also poor. The study found that some genes are upregulated in roots while others in leaves, but their biological significance was not discussed. Different tissues may involve different drought adaptation mechanisms, such as enhanced water absorption in roots and reduced transpiration in leaves, but these functions were not distinguished in the text.
Since the study focuses on DREB, why was ABA treatment not performed? PEG is generally used to simulate osmotic stress but does not fully represent drought-induced stomatal closure and ABA-mediated responses.
The study lacks basic validation. Some of these genes should have been validated in Arabidopsis or other related experiments.
The discussion should be further deepened. The first few paragraphs are almost repetitions of the introduction and results sections.
Root Tissue: The materials section does not mention root samples.
Line 390: 390 What is the basis for these time points? The early response time points, such as 30 min or others were not included, which may result in missing key signal transduction events, such as early ABA induction.
395-340 How was this identified? The study used Arabidopsis DREB genes as a reference to screen for DREB homologous genes in flax. Please provide details on the screening criteria and verify the accuracy of the screening results.
396 The genome used should be cited.
438-451 The instrument used for qRT-PCR should be specified.
448-451 I do not think this approach is appropriate. If these two genes exhibit different expression patterns in different tissues or under different stress conditions, this method could lead to misinterpretation of the data. If one gene has a major effect while the other has low expression, it may obscure the expression of the major-effect gene.
Table S7 The reference gene primers are missing.
Author Response
For research article
|
Response to Reviewer 1 Comments
|
||||
|
1. Summary |
|
|
||
|
We sincerely appreciate your professional comments on our manuscript, which have significantly improved the quality of our work. Below, we provide detailed responses to the points you raised. Our responses are presented in regular font, while changes and additions to the manuscript are highlighted in red text.
|
||||
|
2. Questions for General Evaluation |
Reviewer’s Evaluation |
|
||
|
Does the introduction provide sufficient background and include all relevant references? |
Can be improved |
|||
|
Is the research design appropriate? |
Can be improved |
|||
|
Are the methods adequately described? |
Can be improved |
|||
|
Are the results clearly presented? |
Must be improved |
|||
|
Are the conclusions supported by the results? |
Can be improved |
|||
|
3. Point-by-point response to Comments and Suggestions for Authors |
||||
|
Comments 1: Lines 32-36: Should be more specific in terms of content. |
||||
|
Response 1: Thank you for your comment. We have provided a more detailed description in lines 32-38 on page 1 of our manuscript. The revised sections in the manuscript have been highlighted in red text for easy of reference. |
||||
|
|
||||
|
Comments 2: Lines 80-82: Inappropriate citations, Yuan et al. (2021), Tombuloglu (2020). |
||||
|
Response 2: Thank you for your comment. We sincerely apologize for this oversight. We have revised the inappropriate citation in lines 84-88 on page 3 of the manuscript. The modified content has been highlighted in red text for easy of reference. |
||||
|
|
||||
|
Comments 3: Figure 1: Lacks confidence intervals. |
||||
|
Response 3: We sincerely thank you for pointing this out. Your comment is highly constructive. As we have added a schematic diagram illustrating the functions of DREB transcription factors to the introduction section based on suggestions from other reviewers, this diagram has been labeled as Figure 1 in the revised manuscript. Therefore, the "Figure 1" mentioned in your question has been renumbered as Figure 2 in our revised manuscript. Naturally, we have added the confidence intervals to Figure 2. |
||||
|
|
||||
|
Comments 4: Lines 195-198: The statement should be more cautious. Ks values reflect gene divergence time but cannot be used alone as evidence for WGD events. More background. |
||||
|
Response 4: We sincerely appreciate your comment and agree with your suggestion. Therefore, we have added more background information regarding the relationship between Ks and WGD on lines 199-209 of page 7 in the manuscript. |
||||
|
|
||||
|
Comments 5: Lines 202-204: Why were closely related species of flax not used? |
||||
|
Response 5: We appreciate the reviewer's insightful comment on including closely related flax species in our synteny analysis. In this study, we focused on comparing the DREB genes of flax with those of five other crops to explore evolutionary relationships and functional conservation across a broader phylogenetic range. This approach was chosen to provide a comprehensive perspective on the divergence and conservation of DREB genes in flax relative to other agriculturally important species. However, we acknowledge the value of including closely related flax species for a more detailed evolutionary analysis within the flax lineage. Although such an analysis was not included in the current study due to the limited availability of genomic data for closely related flax species at the time of our analysis, we agree that this represents an important direction for future research. |
||||
|
|
||||
|
Comments 6: Figure 6: The use of raw values makes it difficult to observe trends across samples. The authors should take the log for correction. |
||||
|
Response 6: We thank the reviewer for their comment regarding the use of raw data in Figure 6. As we have added a schematic diagram illustrating the functions of DREB transcription factors, the figure referred to as Figure 6 in your comment has been renumbered as Figure 7 in our revised manuscript. We would like to clarify that the transcriptomic data used in this study were generated in our lab and have been deposited in SRA(accession number: PRJNA505721). The data used in Figure 6 have already been log-transformed (log2) for normalization and visualization purposes. |
||||
|
|
||||
|
Comments 7: Figure 7: This figure is poorly presented. How can relative expression values be negative? Where are the error bars? qRT-PCR should be presented in its conventional format. What does "R48" represent? Why is the green signal missing at "0h" in the figure? The figure lacks a proper caption. The figure caption should be self-explanatory, allowing readers to understand it without referring to the main text. |
||||
|
Response 7: We sincerely thank you for your valuable feedback. Due to the addition of a schematic diagram illustrating the functions of DREB transcription factors, the figure referred to as Figure 7 in your comment has been renumbered as Figure 8 in our revised manuscript. We calculated the relative expression levels of LuDREB genes using the 2−∆∆Ct method, followed by log10 transformation of the data. The transformed data were then used to generate Figure 8. This resulted in negative values for the relative expression levels of the LuDREB genes in Figure 8. R48 represents the samples collected 48h hours after transfer to 1/2 MS liquid medium without PEG supplementation during recovery. Additionally, in calculating the relative expression levels, a control was required as a baseline. We used the expression levels of leaves before PEG treatment as the control. Following log10 transformation, the value of the control became 0, which resulted in the absence of the green signal at "0h" in the figure. We neglected to provide a detailed explanation of this in the Methods section, which might have led to some potential misunderstandings. We sincerely apologize for this oversight. Consequently, we have now included a detailed description of the control selection process in the Methods section on page 18, lines 515-518, to off readers a clearer understanding. Regarding your comment on the inadequacy of the title for Figure 8, we fully agree with your point. We have revised the title of Figure 8 in the manuscript to provide a detailed explanation of the key information, enabling readers to quickly grasp the content without referring to the main text. We believe these improvements will present the data more clearly and minimize potential confusion for readers. Once again, we deeply appreciate your thorough review and constructive suggestions. |
||||
|
|
||||
|
Comments 8: Lines 261-273: The data lacks statistical analysis, yet phrases like "significantly upregulated" and "significantly downregulated" are used. The explanation of the figures is also poor. The study found that some genes are upregulated in roots while others in leaves, but their biological significance was not discussed. Different tissues may involve different drought adaptation mechanisms, such as enhanced water absorption in roots and reduced transpiration in leaves, but these functions were not distinguished in the text. |
||||
|
Response 8: Thank you very much for pointing out these issues. We fully acknowledge your comments. Therefore, we have revised this section of the results by removing potentially misleading terms such as "significantly upregulated" and "significantly downregulated" and replacing them with more precise and appropriate expressions. Additionally, in response to your suggestion to distinguish and discuss the biological significance of the upregulation of LuDREB genes in different tissues, we have extended our analysis and discussion. Relevant literature has also been cited to provide theoretical support for this discussion. The revised section can be found on page 16, lines 409-419 of the manuscript, with changes highlighted in red text for easy identification. |
||||
|
|
||||
|
Comments 9: Since the study focuses on DREB, why was ABA treatment not performed? PEG is generally used to simulate osmotic stress but does not fully represent drought-induced stomatal closure and ABA-mediated responses. |
||||
|
Response 9: We sincerely appreciate the reviewer’s insightful comments on the importance of ABA treatment in studying drought stress responses. In this study, our primary focus was to investigate the role of the LuDREB gene in flax’s response to osmotic stress, a key component of drought conditions. To simulate osmotic stress, we employed PEG treatment, which effectively mimics the dehydration effects of drought stress and allow us to observe the direct impact of water deficit on the expression and function of the LuDREB genes. However, we fully acknowledge the reviewer’s point that PEG treatment alone does not fully capture the complexity of drought-induced responses, especially regarding stomatal closure and ABA-mediated signaling pathways. Indeed, ABA is a critical hormone involved in drought stress responses, and its role in regulating DREB genes and other stress-responsive pathways has been well-documented. Therefore, we plan to incorporate ABA treatment into our follow-up experiments, which will specifically focus on analyzing the expression patterns of LuDREB under ABA-induced stress conditions and its potential interactions with ABA signaling pathways. This will provide deeper insights into the molecular mechanisms through which LuDREB contributes to drought tolerance in flax. However, completing these experiments will require a substantial amount of time. Consequently, we regret being unable to include relevant results regarding ABA treatment in this article. We thank the reviewer for emphasizing this critical aspect and believe that integrating ABA treatment in our future work will substantially broaden the scope and enhance the significance of our findings. |
||||
|
|
||||
|
Comments 10: The study lacks basic validation. Some of these genes should have been validated in Arabidopsis or other related experiments. |
||||
|
Response 10: We sincerely thank the reviewer for their insightful and constructive comments on the necessity of functionally validating the LuDREB genes. We fully acknowledge that functional validation is essential for substantiating the role of LuDREBs in drought stress responses, thereby enhancing both the scientific rigor and practical significance of our manuscript. We have constructed knockout and overexpression vectors for several LuDREB genes, which will subsequently be introduced into Arabidopsis and the flax variety Longya-10. Upon obtaining the transformed plants, we plan to conduct a series of experiments to elucidate the mechanisms by which these genes regulate drought responses. However, completing these experiments is expected to require a substantial amount of time. For instance, obtaining the transformed plants alone can take more than six months. Consequently, we regret being unable to include validation results in this article. |
||||
|
|
||||
|
Comments 11: The discussion should be further deepened. The first few paragraphs are almost repetitions of the introduction and results sections. |
||||
|
Response 11: We sincerely thank you for your valuable suggestions. We fully agree with your comments. Consequently, we have eliminated redundant content in the Discussion section that overlapped with the Introduction. Furthermore, we have streamlined and refined repetitive descriptions of the results to enhance clarity. Simultaneously, we have incorporated more in-depth analyses of the results, particularly focusing on potential interactions between LuDREBs and other transcription factors or signaling pathways. Additionally, we have explicitly addressed the limitations of the current study and outlined prospects for future research. The revised sections are highlighted in red text within the Discussion section of the manuscript. We hope these enhancements will provide readers with deeper insights and inspire further exploration. |
||||
|
|
||||
|
Comments 12: Root Tissue: The materials section does not mention root samples. |
||||
|
Response 12: We sincerely thank you for your valuable comments on our manuscript. Indeed, the Materials and Methods section previously lacked a description of the sampling of root samples, which was an oversight on our part. We have now added a detailed description of the sampling procedures for root samples on page 17, lines 455-457. The revised text has been highlighted in red for easy identification. |
||||
|
|
||||
|
Comments 13: Line 390: 390 What is the basis for these time points? The early response time points, such as 30 min or others were not included, which may result in missing key signal transduction events, such as early ABA induction. |
||||
|
Response 13: Thank you for your valuable comment regarding the selection of time points in our study. The time points of 3h, 6h, 9h, 12h, 24h, 48h, and 72h after PEG treatment were selected based on previous studies investigating drought stress responses in flax and other plant species These studies have demonstrated that these time points are critical for monitoring changes in gene expression associated with drought adaptation. By selecting These time points, we aimed to capture both the mid- and long-term responses of LuDREB genes to drought stress, encompassing their potential roles in osmotic adjustment, stress signaling pathways, and tissue-specific adaptation mechanisms. Relevant references (54 and 76) have been added to justify the selection of sampling time points. [54] Wang, D.; Zeng, Y. Y.; Yang, X. X.; Nie, S. M. Characterization of DREB family genes in Lotus japonicus and LjDREB2B overexpression increased drought tolerance in transgenic Arabidopsis. Bmc Plant Biology 2024, 24, (1). http://doi.org/10.1186/s12870-024-05225-y. [76] Qi, Y. N.; Wang, L. M.; Li, W. J.; Dang, Z.; Xie, Y. P.; Zhao, W.; Zhao, L. R.; Li, W.; Yang, C. X.; Xu, C. M.; Zhang, J. P. Genome-Wide Identification and Expression Analysis of Auxin Response Factor Gene Family in Linum usitatissimum. International Journal of Molecular Sciences 2023, 24, (13). http://doi.org/10.3390/ijms241311006. We acknowledge that incorporating earlier time points, such as 30 minutes or 1 hour, would provide valuable insights into early signal transduction events, including ABA induction and rapid stress responses. However, due to constraints in the experimental design and resource availability, we concentrated on the time points most relevant to the physiological and molecular changes associated with prolonged drought stress. In future studies, we intend to include earlier time points to comprehensively investigate the dynamic expression patterns of LuDREB genes and their roles in early drought stress responses. We appreciate your suggestion, which will undoubtedly enhance the depth and scope of our research. |
||||
|
|
||||
|
Comments 14: 395-340 How was this identified? The study used Arabidopsis DREB genes as a reference to screen for DREB homologous genes in flax. Please provide details on the screening criteria and verify the accuracy of the screening results. |
||||
|
Response 14: We sincerely thank you for this insightful comment. Following the methodology outlined by Emms and Kelly, we employed OrthoFinder 2.5.5 to identify orthologous genes of the Arabidopsis DREB gene family in Longya-10 and pale flax. A detailed explanation of this process is provided in lines 104-105 on page 4 of the manuscript, as well as in section 4.2 of the Methods(page 17, line 469-470). We greatly appreciate your professional feedback, which has significantly contributed to enhancing the quality of our manuscript. |
||||
|
[40] Emms D M, Kelly S. OrthoFinder: phylogenetic orthology inference for comparative genomics. Genome Biol, 2019, 20, (1). http://doi.org/10.1186/s13059-019-1832-y. |
||||
|
|
||||
|
Comments 15: 396 The genome used should be cited. |
||||
|
Response 15: We sincerely thank you for your comment regarding the citation of the genome used in our study. We have now included the relevant reference for the genome source and its database accession numbers in the Methods section (page 17, lines 462-464). The revised text has been highlighted in red within the manuscript. We hope this clarification adequately addresses your concern. |
||||
|
|
||||
|
Comments 16: 438-451 The instrument used for qRT-PCR should be specified. |
||||
|
Response 16: We sincerely appreciate your insightful comment and fully agree with your suggestion. In response, we have now included the specific instrument used for the qRT-PCR experiment in the manuscript on page 18, lines 513-515. To facilitate easy identification, the revised text has been highlighted in red. |
||||
|
|
||||
|
Comments 17: 448-451 I do not think this approach is appropriate. If these two genes exhibit different expression patterns in different tissues or under different stress conditions, this method could lead to misinterpretation of the data. If one gene has a major effect while the other has low expression, it may obscure the expression of the major-effect gene. |
||||
|
Response 17: We sincerely appreciate your insightful comment regarding the potential issues with our approach. The concern raised about the possibility of misinterpretation due to differing expression patterns of the two genes across various tissues or under diverse stress conditions is indeed highly valid. The reason we designed a single primer pair for these gene pairs is that the sequences of the two genes are extremely similar, posing significant challenges in designing gene-specific primers capable of distinguishing between them. We fully acknowledge the limitations of this approach and concur that it may not fully capture the individual contributions of each gene. To address this limitation, we plan to conduct separate functional validation experiments for each gene in our future work. However, this process is time-consuming, and we regret that these results cannot be included in the current manuscript. |
||||
|
|
||||
|
Comments 18: Table S7 The reference gene primers are missing. |
||||
|
Response 18: We sincerely thank you for pointing out the omission of the reference gene primers in Table S7. While the primers for the reference gene were not included in the supplementary table, we did mention in the Methods section on page 18, line 515, that GAPDH was used as the reference gene for normalization. The primers for GAPDH were designed based on the study by Huis et al. (2010), titled "Selection of reference genes for quantitative gene expression normalization in flax (Linum usitatissimum L.)".
|
||||
|
4. Response to Comments on the Quality of English Language |
||||
|
The English is fine and does not require any improvement. |
||||
|
5. Additional clarifications |
||||
|
No additional clarifications are needed. |
||||

Reviewer 2 Report
Comments and Suggestions for Authors
The manuscript focuses on the Dehydration-responsive element-binding (DREB) transcription factors in flax. A total of 59 DREB candidate genes were identified. With their analysis, it provides a clear view of the DREB gene family in flax, which would further help researchers investigate how flax survives from stresses. However, there are still some parts of the manuscript that need to be improved. I would recommend a minor revision for this manuscript before acceptance.
Comments:
-
If LuDREB members refer to proteins, then LuDREB should not be italic.
-
Recommend providing schematic diagram(s) to show the function(s) of the Dehydration-responsive element-binding (DREB).
-
Example as 2.5 title or the other titles: If the title is italic, then the gene names should not be italic.
-
In Figure 7, the title mentions "various tissues." Based on the comparison, it should be specified as 'leaf and root.'
-
In Figure 7, it is recommended to use the same range on the Y-axis. This will easily allow readers to follow and compare.
-
Line 48 mentions "certain gene pairs," including LuDREB16/58, LuDREB17/40, LuDREB18/41, LuDREB46/54, LuDREB12/33, and LuDREB15/38. Please provide more information about the related proteins or the relationships between the subcellular locations of the genes and their encoded proteins.
The English writing can be improved before acceptance.
Author Response
For research article
|
Response to Reviewer 2 Comments
|
||||
|
1. Summary |
|
|
||
|
We sincerely appreciate your professional comments on our manuscript, which have significantly improved the quality of our work. Below, we provide detailed responses to the points you raised. Our responses are presented in regular font, while changes and additions to the manuscript are highlighted in red text.
|
||||
|
2. Questions for General Evaluation |
Reviewer’s Evaluation |
|
||
|
Does the introduction provide sufficient background and include all relevant references? |
Yes |
|||
|
Is the research design appropriate? |
Yes |
|||
|
Are the methods adequately described? |
Yes |
|||
|
Are the results clearly presented? |
Can be improved |
|||
|
Are the conclusions supported by the results? |
Yes |
|||
|
3. Point-by-point response to Comments and Suggestions for Authors |
||||
|
Comments 1: If LuDREB members refer to proteins, then LuDREB should not be italic. |
||||
|
Response 1: Thank you for raising this point and fully. agree with your comment. Accordingly, we have made the following modifications in the revised manuscript: In lines 21, 159, 171, and 329 of the manuscript, the italic formatting for LuDREB has been removed when it refers to the protein. The modified sections are highlighted in red for easy identification. |
||||
|
|
||||
|
Comments 2: Recommend providing schematic diagram(s) to show the function(s) of the Dehydration-responsive element-binding (DREB). |
||||
|
Response 2: We sincerely thank you for your valuable suggestions regarding our manuscript. In response, we have added a schematic diagram to the Introduction section, illustrating the functions of DREB transcription factors. This diagram is presented as Figure 1 in the revised version of the manuscript.
|
||||
|
Comments 3: Example as 2.5 title or the other titles: If the title is italic, then the gene names should not be italic. |
||||
|
Response 3: We thank you for your suggestion and fully agree with this comment. Accordingly, we have updated the gene names in the title to a non-italicized format in the revised manuscript. The modified section has been highlighted in red for easy identification. |
||||
|
|
||||
|
Comments 4: In Figure 7, the title mentions "various tissues." Based on the comparison, it should be specified as 'leaf and root.' |
||||
|
Response 4: We sincerely appreciate your valuable comment. We fully agree with this suggestion. In the revised manuscript, a schematic diagram illustrating the function of DREB has been added to the introduction section, which resulted in the renumbering of the figures. Consequently, the figure previously referred to as Figure 7 has now been renumbered as Figure 8. Following your recommendation, we have revised the title of Figure 7 (now Figure 8) to "The expression profiles of LuDREB genes in leaves and roots detected by qRT-PCR under PEG treatment." The modified section has been highlighted in red for easy identification. |
||||
|
|
||||
|
Comments 5: In Figure 7, it is recommended to use the same range on the Y-axis. This will easily allow readers to follow and compare. |
||||
|
Response 5: We sincerely appreciate your valuable comment. We fully agree with this suggestion. In the revised manuscript, a schematic diagram illustrating the function of DREB has been added to the introduction section, which resulted in the renumbering of the figures. Consequently, the figure previously referred to as Figure 7 has now been renumbered as Figure 8. Following your recommendation, we have standardized the Y-axis in Figure 7 (now Figure 8) to a uniform range of -4 to 3. However, it is worth noting that the expression levels of certain genes consistently increased or decreased over the treatment period, such as LuDREB1, LuDREB11/51, and LuDREB28. For these genes, the Y-axis range of the bar graphs illustrating expression level changes has been adjusted to -4 to 0 or 0 to 3, which may enhance the clarity of the data presentation. Additionally, for genes with less pronounced changes in expression levels, we have appropriately narrowed the Y-axis range in their respective bar graphs to enable a more precise observation of the expression trends, as observed with LuDREB5/32 and LuDREB33. These specific modifications to the gene images are designed to enhance the reader's experience while ensuring that the ability to track and compare results is not compromised. |
||||
|
|
||||
|
Comments 6: Line 48 mentions "certain gene pairs," including LuDREB16/58, LuDREB17/40, LuDREB18/41, LuDREB46/54, LuDREB12/33, and LuDREB15/38. Please provide more information about the related proteins or the relationships between the subcellular locations of the genes and their encoded proteins. |
||||
|
Response 6: We sincerely thank you for your insightful comment regarding the gene pairs mentioned, including LuDREB16/58, LuDREB17/40, LuDREB18/41, LuDREB46/54, LuDREB12/33, and LuDREB15/38. Although detailed information and subcellular localization predictions for all LuDREB proteins have been provided in the supplementary tables of the manuscript, we acknowledge that the precise subcellular localization of these specific gene pairs and the relationship between their localization and the functions of their encoded proteins require warrant further investigation. We plan to conduct more detailed investigations into these aspects in our future work. However, given the time-consuming nature of such research, we are unable to provide more comprehensive information on these genes in the current manuscript. We deeply regret this limitation and sincerely appreciate your understanding.
|
||||
|
4. Response to Comments on the Quality of English Language |
||||
|
Point 1: The English writing can be improved before acceptance. |
||||
|
Response 1: We sincerely thank you for your meticulous review of our manuscript and your insightful and constructive suggestions for enhancing the English writing. Before submission, we had already undertaken efforts to enhance the English quality of our manuscript. The text was carefully refined by a professional English editing service to ensure clarity and readability. In response to your comments, we have conducted a thorough re-examination of the manuscript and made additional revisions to certain expressions that could be further improved for greater clarity. The revised manuscript has now been updated, and we hope that these linguistic improvements meet your expectations. We remain open to any further feedback to continuously elevate the quality of our work. |
||||
|
5. Additional clarifications |
||||
|
No additional clarifications are needed. |
||||

Reviewer 3 Report
Comments and Suggestions for Authors
A review of the manuscript: Genome-wide identification and expression profiling of dehydration-responsive element-binding family genes in flax (Linum usitatissimum L.).
The present study focuses on identifying LuDREB genes in Longya-10, their molecular characterisation, phylogenetic analysis, chromosomal localization, gene duplication analysis, cis regulatory elements and expression patterns in different tissues under drought stress. This study aims to gain a more in-depth understanding of the role these genes play in responding to abiotic stresses, with a particular focus on drought, and to ascertain their potential importance in the development of flax.
The research presented in the manuscript is of a high scientific standard and follows current trends in plant genomics and abiotic stress biology. The authors have employed a comprehensive array of contemporary bioinformatics and experimental methodologies.
In certain sections, the authors' depiction of the results is overly technical, which may impede readers' swift comprehension of their biological implications. For instance, in section 2.6 ("Expression analysis of LuDREB genes ..."), the authors report that LuDREB23 was significantly up-regulated at all time points in leaves and roots. Yet, they do not indicate the potential implications of this finding for gene function. Integrating interpretation with data description could enhance the readability and appeal of the text.
The discussion could be enriched by considering potential interactions of LuDREB with other transcription factors or signalling pathways, which is essential in the context of the complex stress response.
The authors should provide more explicit discussion of the study's limitations, including the absence of field condition data and the lack of functional analysis in transgenic models. They should also propose specific directions for further research.
Author Response
For research article
|
Response to Reviewer 3 Comments
|
||||
|
1. Summary |
|
|
||
|
We sincerely appreciate your professional comments on our manuscript, which have significantly improved the quality of our work. Below, we provide detailed responses to the points you raised. Our responses are presented in regular font, while changes and additions to the manuscript are highlighted in red text.
|
||||
|
2. Questions for General Evaluation |
Reviewer’s Evaluation |
|
||
|
Does the introduction provide sufficient background and include all relevant references? |
Yes |
|||
|
Is the research design appropriate? |
Yes |
|||
|
Are the methods adequately described? |
Yes |
|||
|
Are the results clearly presented? |
Can be improved |
|||
|
Are the conclusions supported by the results? |
Yes |
|||
|
3. Point-by-point response to Comments and Suggestions for Authors |
||||
|
Comments 1: In certain sections, the authors' depiction of the results is overly technical, which may impede readers' swift comprehension of their biological implications. For instance, in section 2.6 ("Expression analysis of LuDREB genes ..."), the authors report that LuDREB23 was significantly up-regulated at all time points in leaves and roots. Yet, they do not indicate the potential implications of this finding for gene function. Integrating interpretation with data description could enhance the readability and appeal of the text. |
||||
|
Response 1: Thank you for your meticulous review and constructive feedback on our manuscript. We have revised Section 2.6 in accordance with your suggestions and integrated discussions on the biological implications of the up- and down-regulation of LuDREB genes while presenting the experimental results. These revisions can be found on page 12, lines 276–290 of the manuscript, where the updated text is highlighted in red. We believe that these additional discussions not only facilitate a deeper understanding of the experimental results for readers but also more clearly elucidate the potential functions of LuDREB genes in plant stress responses. We hope these revisions align with your expectations and further enhance the readability and scientific rigor of the manuscript. |
||||
|
|
||||
|
Comments 2: The discussion could be enriched by considering potential interactions of LuDREB with other transcription factors or signalling pathways, which is essential in the context of the complex stress response. |
||||
|
Response 2: We sincerely appreciate your professional suggestions. In response to your comments, we have provided an in-depth discussion regarding the potential interactions of LuDREBs with other transcription factors or signaling pathways in lines 376-397 on page 15 of the manuscript. |
||||
|
|
||||
|
Comments 3: The authors should provide more explicit discussion of the study's limitations, including the absence of field condition data and the lack of functional analysis in transgenic models. They should also propose specific directions for further research. |
||||
|
Response 3: We sincerely appreciate your valuable suggestions and fully agree with your perspective. Accordingly, we have provided a detailed discussion regarding the limitations of this study and specific directions for future research in lines 436-445 on page 16 of the manuscript. This discussion is instrumental in enabling us to identify the shortcomings of our current work and guiding our next steps. We are deeply grateful for your professional feedback, which has played a pivotal role in enhancing the quality of our manuscript. |
||||
|
|
||||
|
4. Response to Comments on the Quality of English Language |
||||
|
The English is fine and does not require any improvement. |
||||
|
5. Additional clarifications |
||||
|
No additional clarifications are needed. |
||||

Round 2
Reviewer 1 Report
Comments and Suggestions for Authors
The authors have addressed most of the issues, but there are still some problems.
“Comment 6: Figure 6: The use of raw values makes it difficult to observe trends across samples. The authors should take the log for correction.
Response 6: We thank the reviewer for their comment regarding the use of raw data in Figure 6. As we have added a schematic diagram illustrating the functions of DREB transcription factors, the figure referred to as Figure 6 in your comment has been renumbered as Figure 7 in our revised manuscript. We would like to clarify that the transcriptomic data used in this study were generated in our lab and have been deposited in SRA (accession number: PRJNA505721). The data used in Figure 6 have already been log-transformed (log2) for normalization and visualization purposes.”
Are you sure the values were log2-transformed? The color bar shows a maximum value of 120, which would imply that log2(FPKM+x) = 120. That would correspond to an FPKM value of 2^120, which is absolutely impossible.
In addition, the figure legend of Figure 8 should include the phrase "using the 2^−ΔΔCt method", followed by a note that a logarithmic transformation (log10) was applied afterward.
Author Response
|
Response to Reviewer 1 Comments (Second Round)
|
||||
|
1. Summary |
|
|
||
|
We sincerely appreciate your additional professional comments on our manuscript, which have further significantly improved the quality of our work. Below, we provide detailed responses to the points you raised. Our responses are presented in regular font, while changes and additions to the manuscript are highlighted in red text.
|
||||
|
2. Questions for General Evaluation |
Reviewer’s Evaluation |
|
||
|
Does the introduction provide sufficient background and include all relevant references? |
Can be improved |
|||
|
Is the research design appropriate? |
Can be improved |
|||
|
Are the methods adequately described? |
Can be improved |
|||
|
Are the results clearly presented? |
Can be improved |
|||
|
Are the conclusions supported by the results? |
Can be improved |
|||
|
3. Point-by-point response to Comments and Suggestions for Authors |
||||
|
Comments 1: Are you sure the values were log2-transformed? The color bar shows a maximum value of 120, which would imply that log2(FPKM+x) = 120. That would correspond to an FPKM value of 2^120, which is absolutely impossible. |
||||
|
Response 1: We sincerely thank you for pointing out the issue with the data in Figure 7 of our manuscript. Upon receiving your comment, we immediately re-examined and recalculated our raw data, and we identified a significant error: the heatmap was generated using the raw data instead of the log2-transformed data. This was a serious oversight in our work, and we deeply apologize for this mistake. At the same time, we are extremely grateful for your professionalism and rigor, which helped us identify and correct this critical error. We have now recreated Figure 7 using the log2-transformed data and have explicitly stated in the figure caption that the data were log2-transformed. The revised content can be found on page 11, line 272 of the updated manuscript. |
||||
|
|
||||
|
Comments 2: In addition, the figure legend of Figure 8 should include the phrase "using the 2^−ΔΔCt method", followed by a note that a logarithmic transformation (log10) was applied afterward. |
||||
|
Response 2: We sincerely appreciate your professional suggestion. We have updated the caption of Figure 8 to include the following statement: "Relative expression levels were calculated using the 2−ΔΔCt method, followed by logarithmic transformation (log10)." The revised content can be found on page 13, lines 298-299 of the manuscript. |
||||
|
4. Response to Comments on the Quality of English Language |
||||
|
The English is fine and does not require any improvement. |
||||
|
5. Additional clarifications |
||||
|
No additional clarifications are needed. |
||||
